# Hydrogen Peroxide Ameliorates the Adversities of Drought Stress during Germination and Seedling Growth in Sorghum (*Sorghum bicolor* L.)

Ki Eun Song [1,†], He Rin Hwang [1,†], e Sun Hee Hong [2], Petr Konvalina [3], Woo Jin Jun [4], Jin Woong Jung [5] and Sangin Shim [6,*]

1 Department of Agronomy, Gyeongsang National University, Jinju 52828, Republic of Korea
2 Department of Plant Life Science, Hankyong National University, Ansung 17579, Republic of Korea
3 Faculty of Agriculture and Technology, University of South Bohemia in České Budějovice, Studentská 1668, 37005 České Budějovice, Czech Republic
4 Department of Food and Nutrition, Chonnam National University, Gwangju 61186, Republic of Korea
5 Department of Biological Science, Dong-A University, Busan 49315, Republic of Korea
6 Department of Agronomy, Institute of Agriculture and Life Sciences, Gyeongsang National University, Jinju 52828, Republic of Korea
* Correspondence: sishim@gnu.ac.kr; Tel.: +82-55-772-1873
† These authors contributed equally to this work as co-first authors.

**Abstract:** Proper seed germination is important for seedling establishment and growth in fields under drought caused by climate change. In the present study, the beneficial effects of hydrogen peroxide on seed germination were investigated by proteome analysis. Sorghum seeds were subjected to drought stress adjusted to the various water potentials of 0, −0.2, and −0.5 MPa using polyethylene glycol (PEG) 6000 and treated with hydrogen peroxide at different concentrations (0, 10, 20, 50, and 100 mM). Germination percentage and seedling growth were determined at 6 days after imbibition, and proteins from embryos were analyzed. As a result of the study, it was found that the reduced germination percentage and seedling growth under drought stress were alleviated by hydrogen peroxide treatment. Proteins induced in hydrogen peroxide-treated embryos included glycolysis-related enzymes (25%) and stress-defense proteins (30%). Among the glycolysis-related enzymes, fructokinase-1 was higher only under drought and 0 mM $H_2O_2$ treatment, and phosphoglycerate kinase was higher than control under drought and 100 mM $H_2O_2$ treatment. Hydrogen peroxide treatment also increased the expression of antioxidant enzymes related to stress defense. The results that hydrogen peroxide treatment increases germination rate and seedling growth by increasing energy production and defense activity suggest a practical application of $H_2O_2$ at 100 mM for reducing the drought stress in sorghum.

**Keywords:** drought; hydrogen peroxide; proteome; seed germination; *Sorghum bicolor*



## 1. Introduction

Sorghum (*Sorghum bicolor*) is a representative $C_4$ plant, and it is a crop with increasing production around the world due to its high use as a bioethanol material and feed crop. Sorghum is often grown in arid and semi-arid regions, where drought is the main limiting factor for crop cultivation. Bayu et al. [1] reported that the germination and seedling growth of sorghum were inhibited at 60% field capacity (weak drought) and 40% field capacity (strong drought). Seed germination of sorghum is reduced soil at a water potential of 0 MPa to −0.8 MPa due to the inhibition of radicle cell division and elongation [2,3]. Drought tolerance of crops is important in agricultural fields where the threat of drought in relation to the field emergence of plants is increased by climate change. The strong drought tolerance of sorghum increases the likelihood of growing this crop in water-scarce conditions, and it

also shows the need to study the drought resistance of this plant. Sorghum is often used as an experimental crop to study the physiological mechanisms of drought tolerance [4,5].

The seed stores nutrients used for the growth of seedlings [6,7], as well as for emerging radicle and plumule from embryos through digestion and assimilation of the stored nutrients [8]. Germination is a sensitive period to abiotic stress [9]. Drought stress imposed at the beginning of germination harms not only germination but also seedling growth due to the hydrolysis of stored nutrients limited by drought stress [10]. The osmotic regulator is used for controlling the water potential in germination experiments under drought stress. Among the osmotic substances, polyethylene glycol (PEG) is often used in germination experiments because it successfully mimics drought stress and easily controls the water potential [11].

The levels of reactive oxygen species (ROS) such as hydrogen peroxide, hydroxyl radicals, and superoxide radicals change in embryos during germination and influence the appearance of radicle and plumule [12–15]. In particular, hydrogen peroxide had a positive effect on seedling growth by promoting the germination of dormant seeds [16–18] and survival rate [19] of seedlings, and it lowered the inhibitory effect of salt stress on germination when sunflower seeds were treated with hydrogen peroxide [20]. Hydrogen peroxide treatment increases seedling growth via interaction with plant hormones related to the proteins that participate in crop growth, cell signaling, and cycle regulation [21]. Although there are many results of the beneficial effect of hydrogen peroxide on germination and dormancy breaking [17,19,21,22], there are fewer reports on germination under water stress conditions. Although ROS play an important role in germination [13,15,18–20], there is little study on stress mitigation by hydrogen peroxide during germination under water stress.

Proteomic analysis can compare proteins and obtain information on individual proteins related to physiological functions during germination [23–25]. It is possible to simultaneously classify the pattern of proteins that occurs within the complex germination process in various plants such as *Arabidopsis* [26], rice [27,28], tomato [29], and soybean [30,31]. Proteomic analyses to identify the effect of chemical treatment that increases stress or stress-tolerant crop under environmental stress have been conducted [32,33], but there are few studies on protein analysis according to hydrogen peroxide treatment in seed germination.

The purpose of this study was to clarify the effect of hydrogen peroxide treatment on germination under water stress conditions, becoming a frequent event critical to the emergence and seedling establishment at the beginning of growing in sorghum fields. The beneficial effects of hydrogen peroxide can be applied to improve the drought tolerance in arable fields of not only arid or semi-arid regions but also presumably dry regions threatened by climate change.

## 2. Materials and Methods

### 2.1. Experimental Materials and Germination Conditions

Seeds of sorghum (cv. Hwangeumchal) were subjected to solutions with different water potentials adjusted to 0 (wet), −0.2 (weak water stress), and −0.5 MPa (strong water stress) using PEG 6000 [34], along with hydrogen peroxide concentrations in the medium of 0, 10, 20, 50, and 100 mM. Seeds were placed on filter paper (Whatman No. 1) lined in a 15 cm plastic petri dish containing 3 mL of medium. All treatment combinations were replicated four times. One hundred seeds of each treatment per replication were subjected to different water potentials and hydrogen peroxide concentrations under dark conditions at a constant temperature of 25 °C in a growth chamber. The seeds observed with a protrusion of radicle by at least 2 mm were considered germinated seeds. The germination test was performed for 120 h, and germinated seeds were counted every 6 h. The average germination percentage and the length of the radicle and plumule were measured on the last day of the germination test. The imbibition percentage was measured at 24 h after treatment as follows: imbibition (%) = (FW − DW)/DW × 100, where FW is the fresh weight of the seeds 24 h after the treatment, and DW is the dry weight of the

seeds at 0 h [35]. The embryos treated with hydrogen peroxide were dissected with a razor blade, collected in liquid nitrogen, and stored at –80 °C for the analysis of oxygen radical absorbance capacity (ORAC) and protein analysis (2DE).

### 2.2. Antioxidant Capacity (ORAC) Assay

After 24 h of imbibition, the embryos were cut out with a razor blade and stored at −80 °C until use. The antioxidant activity of embryos was measured by ORAC assay according to the previously described method [36]. The antioxidant capacity of a substance can be directly estimated by comparison to the standard curve of Trolox.

### 2.3. Protein Extraction and Two-Dimensional Electrophoresis

A frozen sample powder of embryos collected at 6 days after imbibition was put into a 15 mL tube, and the protein was extracted using the method of Wang et al. [37]. After extraction, lysis buffer (7 M Urea, 2 M Thiourea, 4% CHAPS, 40 mM DTT) was added to the dried protein pellet and reacted for 1 h at room temperature; after centrifugation, the supernatant was transferred to a 1.5 mL tube and stored at −80 °C until use. The extracted protein was quantified using the Bradford method [38], and the proteins (900 μg) were rehydrated and focused using a Protean IEF Cell (Bio-Rad) according to the method of Kim et al. [39]. After equilibrating the focused protein, the strip was placed on a 12.5% SDS gel (185 × 200 × 1.0 mm) and subjected to electrophoresis using a Protean II XI electrophoresis device (Bio-Rad). The proteins were stained for 24 h in Coomassie brilliant blue G-250 (Genomic Base) solution.

### 2.4. Protein Identification Using Mass Spectrometry (MALDI-TOF/TOF Mass Spectrometry)

Protein spots stained with Coomassie brilliant blue were visualized using a digital scanner (Epson Perfection V800 Photo, Japan). For the accuracy of the data, the 2-DE analysis was performed through three replications. The gel image was analyzed using PDQuest version 7.2.0 software (Bio-Rad). After the image analysis, spots with statistically significant differences ($p < 0.05$, Student's *t*-test) of more than 1.5 times were used for protein analysis, and then different protein spots were separated from the gel. The gel piece containing stained protein was separated according to the method of Liu et al. [40]. In-gel digestion and tryptic peptide extraction were performed using the method of Shevchenko et al. [41]. The gel pieces were repeatedly washed and destained using water and acetonitrile at 20 °C. The proteins in the gel piece were reduced with 10 mM DTT in 100 mM $NH_4HCO_3$ at 56 °C, and then incubated with 55 mM iodoacetamide in 100 mM $NH_4HCO_3$ for 30 min at 20 °C. For digestion, the gel pieces were rehydrated in 50 mM $NH_4HCO_3$ with 10 ng of trypsin on ice for 45 min and then incubated overnight at 37 °C. After digestion, peptides were sequentially extracted using 5 μL of 5% (*v/v*) trifluoroacetic acid (TFA) followed by 50 μL of 50% (*v/v*) acetonitrile with 2.5% (*v/v*) TFA. Each sample was sonicated for 5 min before removing the supernatant. Peptides were desalted using a column of POROS R2 resin (AB Sciex, Foster City, CA, USA), and then the samples were spotted on a MALDI target plate. Samples were then analyzed using an AB Sciex 4800 Plus MALDI-TOF/TOF mass spectrometer (Applied Biosystems, Franklin Lakes, NJ, USA). The peptide mass fingerprints (PMFs) obtained were searched against sorghum in the Swissprot and/or NCBI databases using the MS-Fit program (http://prospector.ucsf.edu, accessed on 15 October 2020), and PMFs were identified by first searching in the Swissprot and NCBI databases. The possible functions of identified proteins were determined according to the information from UniProt on sorghum (http://www.uniprot.org, accessed on 15 October 2020).

### 2.5. Statistical Analysis

Data were subjected to analysis of variance (ANOVA) using SAS software (ver. 9.4. SAS Inst., Cary, NC, USA). One-way ANOVA was applied to compare the means between different treatments. When the *p*-value was <0.05, Tukey's HSD test was applied for

pairwise analysis. Each treatment value was presented as the arithmetic mean $\pm$ standard error (n = 4) in graphs.

## 3. Results

### 3.1. Hydrogen Peroxide Effects on Seed Germination under Drought Stress

The germination percentage was more than 70% in all hydrogen peroxide concentrations examined until 120 h in the control condition (0 MPa), but there was no difference between the concentrations of hydrogen peroxide (Figure 1). However, the germination percentage by the hydrogen peroxide concentration increased after 36 h in the weak water stress condition ($-0.2$ MPa) and 60 h after the commencement of germination in the strong drought stress condition ($-0.5$ MPa) (Figure 1). Hydrogen peroxide at low concentrations (10 mM and 20 mM) showed a lower germination rate than the control at $-0.2$ MPa, but hydrogen peroxide of 50 mM and 100 mM increased the germination rate. Under strong water stress ($-0.5$ MPa), hydrogen peroxide promoted germination at all concentrations treated. After 120 h, the average germination rate was 4.0% higher in weak water stress conditions and 20.0% higher in strong water stress conditions in 100 mM $H_2O_2$ treatment as compared to the 0 mM $H_2O_2$ treatment.

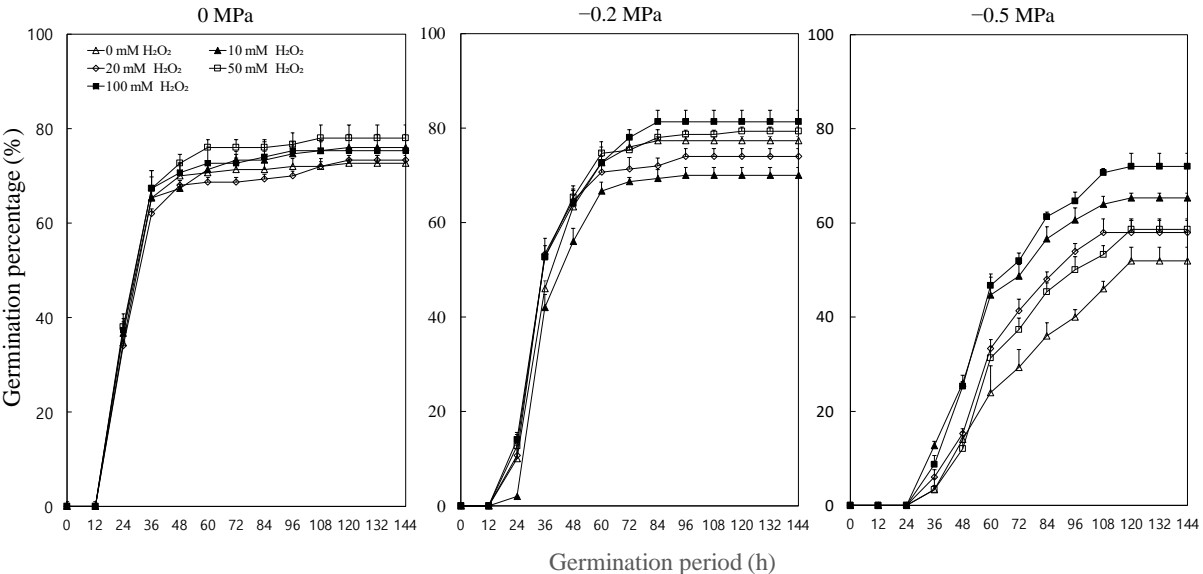

**Figure 1.** Effect of hydrogen peroxide on germination of sorghum seeds under different drought conditions imposed with various water potentials of 0, $-0.2$, and $-0.5$ MPa controlled with PEG 6000. Hydrogen peroxide was treated at the rate of 0, 10, 20, 50, and 100 mM in PEG solution. All values are the means $\pm$ SE of three replicates of 30 seeds each. The protrusion of a 2 mm radicle was used as the criterion for the completion of germination. All values are the means $\pm$ SE of four replicates of 30 seeds each.

### 3.2. Hydrogen Peroxide Effects on Seed Imbibition

The imbibition of seeds and the antioxidant ability of the embryo were investigated in 100 mM $H_2O_2$ at which the germination percentage and seedling length were greater under strong drought stress conditions. Imbibition percentage was the highest at 40.1% in seeds with 0 mM $H_2O_2$ treatment under optimum moisture conditions, but the imbibition between seeds treated with 0 mM $H_2O_2$ and 100 mM $H_2O_2$ was similar under strong drought stress conditions (Figure 2).

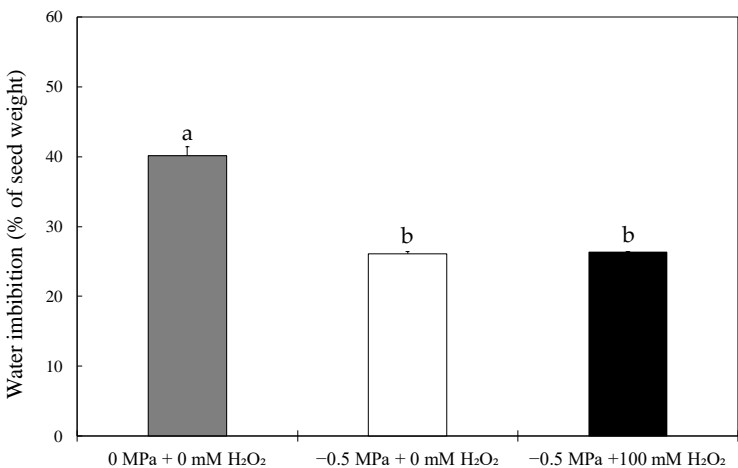

**Figure 2.** Effect of hydrogen peroxide treatment on the water absorption percentage of seeds at 24 h after imbibition under different water potentials. Seeds were imbibed with distilled water (0 MPa) without $H_2O_2$ and PEG solution ($-0.5$ MPa) with or without 100 mM $H_2O_2$. Columns with the same letters are not significantly different between treatments by Tukey's test ($p < 0.05$).

*3.3. Hydrogen Peroxide-Induced Seedling Growth Response under Drought Stress*

At 120 h after imbibition, the sorghum seedlings germinated and grown under each moisture condition were divided into plumule and radicle, and the length was measured (Figure 3). As a result of comparing the lengths of plumules and radicles according to the concentration of hydrogen peroxide under optimum moisture conditions (0 MPa), the lengths of plumules and radicles without hydrogen peroxide treatment were 27.9 cm and 38.2 cm, respectively. However, under weak drought stress conditions, the length of plumules and radicles treated with 20 mM $H_2O_2$ was 7.7 cm and 10.4 cm, respectively. Plumules were longer in the 10 mM $H_2O_2$ treatment (1.7 cm) and 100 mM $H_2O_2$ treatment (1.6 cm), and the radicle was the longest at 2.4 cm in the 10 mM $H_2O_2$ treatment and 100 mM $H_2O_2$ treatment under strong drought stress conditions (Figure 3).

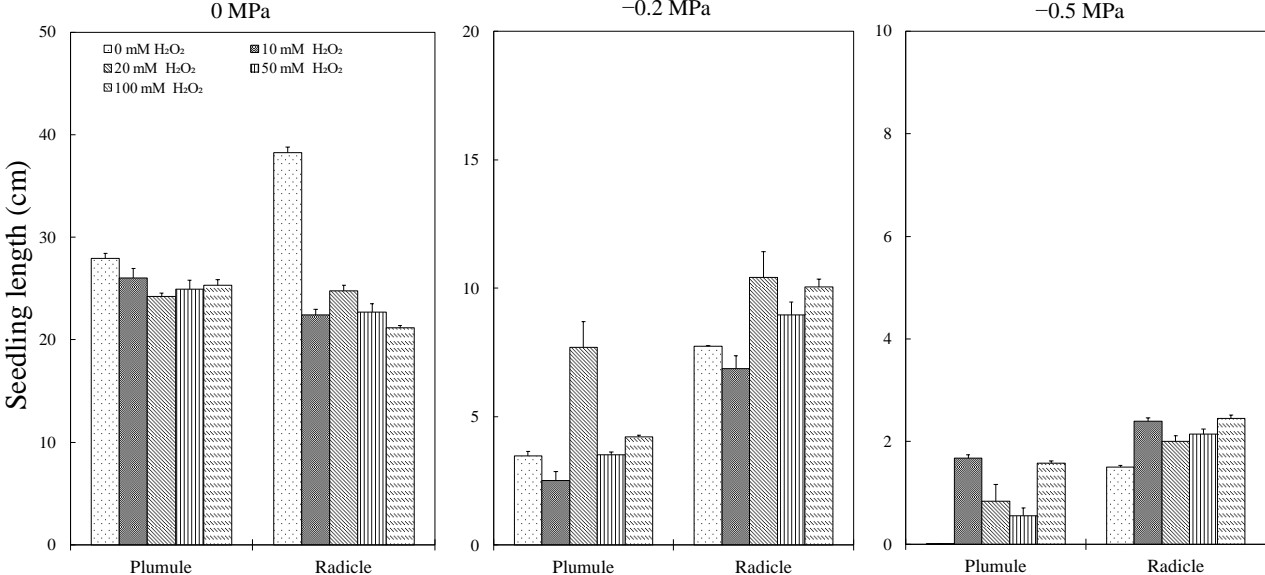

**Figure 3.** Effects of hydrogen peroxide on the growth of plumule and radicle from the embryonic axis of sorghum seed under various water potentials. Plumule and radicle lengths were measured after 120 h of imbibition. All values are the means ± SE of three replicates of 30 seeds each. Columns with the same letters are not significantly different between treatments according to Tukey's test ($p < 0.05$).

### 3.4. Antioxidant Activity under Drought Stress

Antioxidant activity (ORAC) of the embryos treated with 0 mM $H_2O_2$ was higher by 496.8 µM Trolox·$g^{-1}$ FW than embryos treated with 100 mM $H_2O_2$ under strong drought stress conditions. However, The ORAC values were similar between the embryos treated with 0 mM $H_2O_2$ under optimum conditions and embryos treated with 100 mM $H_2O_2$ under strong drought stress conditions (Figure 4).

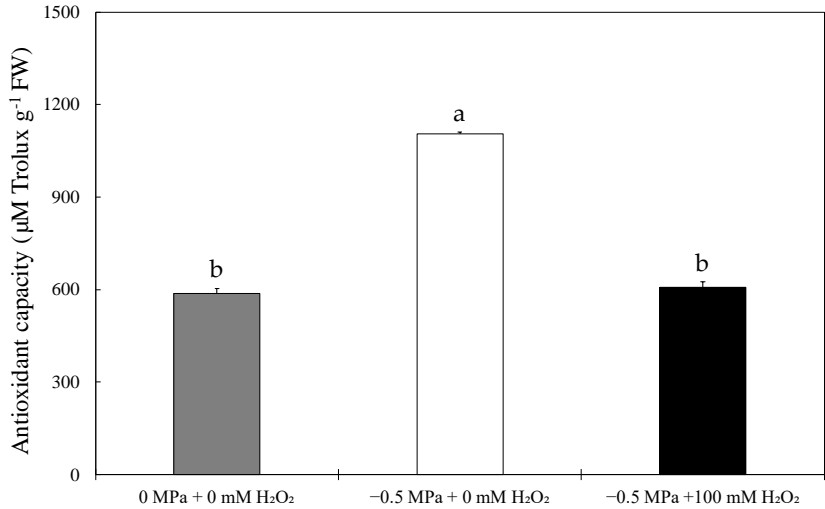

**Figure 4.** Effect of hydrogen peroxide treatment on the antioxidant activity of seeds at 24 h after imbibition under different water potentials. Antioxidant activity is expressed as micromoles of Trolox equivalent per gram based on the fresh weight. Seeds were imbibed with distilled water (0 MPa) without $H_2O_2$ and PEG solution (−0.5 MPa) with or without 100 mM $H_2O_2$. Columns with the same letters are not significantly different between treatments according to Tukey's test ($p < 0.05$).

### 3.5. Identification of Differentially Abundant Proteins and Their Functional Classification

The numbers of proteins detected in 2-DE gel were 403, 348 and 415 for control (0 MPa + 0 mM $H_2O_2$), drought without $H_2O_2$ (−0.5 MPa + 0 mM $H_2O_2$), and drought with $H_2O_2$ (−0.5 MPa + 100 mM $H_2O_2$), respectively. Up- and downregulation occurred differently according to $H_2O_2$ concentrations (Figure 5). To know the effect of hydrogen peroxide treatment on sorghum seed proteins during germination under drought conditions, a total of 40 proteins from the embryos were identified by MALDI-TOF/TOF MS analysis (Figure 5; Table 1). As a result of classifying the identified proteins, stress defense (30%), energy metabolism (25%), amino-acid metabolism (15%), storage protein (12.5%), cell-wall and plant cytoskeletal structural proteins (10%), and other proteins (7.5%) appeared (Figure 6). Among the most identified stress defense proteins, spots 5, 6, 9, 10, 14, 17, 29, and 31 were high in the embryos treated with strong drought and 100 mM $H_2O_2$ and spots 26, 27, 28, and 41 were high in optimal conditions and 0 mM $H_2O_2$ treatment. Specifically, the VOC domain-containing protein (spot 2) was strongly induced in the embryos with 0 mM $H_2O_2$ treatment under strong drought conditions (Figure 5; Table 1). Protein spots 18, 21, 24, 34, 35, 36, 38, and 40 related to energy metabolism showed a high intensity under optimum moisture conditions with 0 mM $H_2O_2$ treatment and the strong drought with 100 mM $H_2O_2$ treatment. Fructokinase-1 (spot 16), however, was higher in the embryos treated with 0 mM $H_2O_2$ under strong drought conditions.

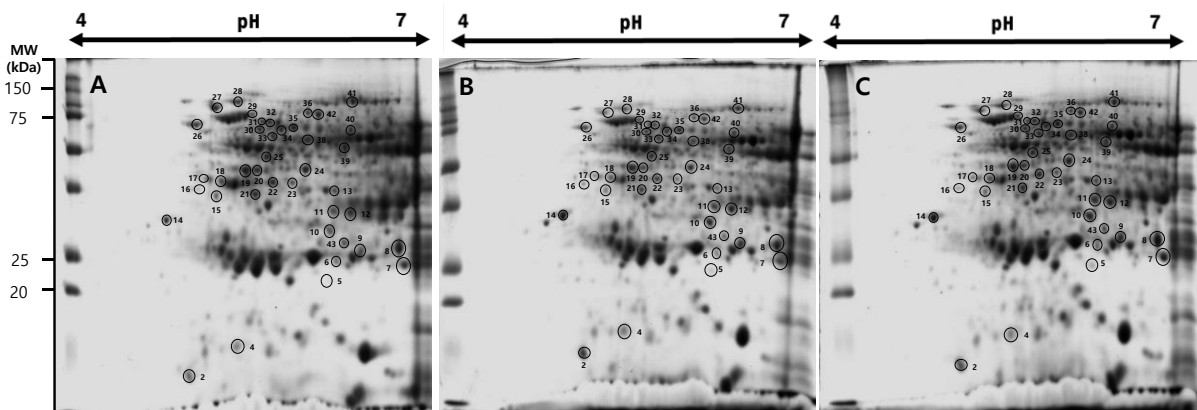

**Figure 5.** The 2DE-images of seed proteome extracted at 24 h after germination. Protein extracts of embryos from control (**A**, 0 MPa + 0 mM $H_2O_2$) and drought stress-treated sorghum seeds (**B**, −0.5 MPa + 0 mM $H_2O_2$), and drought stress and hydrogen peroxide treated seeds (**C**, −0.5 MPa + 100 mM $H_2O_2$) were separated by 2DE (pH 4–7, 12.5% SDS-PAGE).

**Table 1.** Differentially expressed proteins identified in sorghum seed embryos at 24 h after treatment.

| Spot No. | Accession No. | Protein Name | Score [a] | Mr [b] (kD) | pI [c] | SC [d] | Category | Fold Change (−0.5 MPa + 0 mM $H_2O_2$) [e] | Fold Change (−0.5 MPa + 100 mM $H_2O_2$) [f] |
|---|---|---|---|---|---|---|---|---|---|
| 2 | C5WPE0 | VOC domain-containing protein | 107 | 15,336 | 5.00 | 66 | Stress defense | +1.58 | +1.69 |
| 5 | XP_002439542 | Superoxide dismutase [Mn] 3.4, mitochondrial | 189 | 25,341 | 6.71 | 27 | Stress defense | +2.04 | +5.56 |
| 6 | AAA33469 | Glutathione S-transferase I | 172 | 24,003 | 5.44 | 34 | Stress defense | −1.24 | +1.42 |
| 9 | XP_002461098 | 1-Cys peroxiredoxin PER 1 | 133 | 24,283 | 6.17 | 46 | Stress defense | +1.74 | +1.72 |
| 10 | XP_002463881 | Late embryogenesis abundant protein D-34 | 296 | 27,709 | 5.87 | 48 | Stress defense | +1.12 | +1.35 |
| 14 | XP_002465829 | Late embryogenesis abundant protein D-34 | 123 | 27,937 | 4.77 | 32 | Stress defense | +2.29 | +2.32 |
| 17 | B6T887 | Salt tolerance protein | 219 | 35,239 | 4.90 | 37 | Stress defense | +1.33 | +1.87 |
| 26 | ABP88739 | Protein disulfide isomerase | 182 | 56,921 | 5.01 | 20 | Stress defense | −1.68 | +2.27 |
| 27 | G7KWU8 | Heat-shock cognate 70 kDa protein | 96 | 71,260 | 5.07 | 23 | Stress defense | −1.83 | −1.61 |
| 28 | XP_002465468 | Heat-shock cognate 70 kDa protein | 459 | 71,756 | 5.19 | 54 | Stress defense | −2.77 | −1.69 |
| 29 | AAA33450 | Chaperonin CPN60-1, mitochondrial | 161 | 61,484 | 5.68 | 41 | Stress defense | +1.06 | +1.41 |
| 41 | C5Z1B4 | Uncharacterized protein | 321 | 101,354 | 5.75 | 34 | Stress defense | −2.16 | +1.14 |
| 16 | Q6XZ79 | Fructokinase-1 | 569 | 34,840 | 4.87 | 64 | Energy metabolism | +2.20 | +1.94 |
| 18 | Q9XGC6 | Adenosine kinase | 204 | 36,465 | 5.23 | 45 | Energy metabolism | −1.65 | +1.02 |
| 21 | A0A1Z5RCS5 | Uncharacterized protein | 126 | 33,525 | 5.43 | 53 | Energy metabolism | −1.68 | +1.15 |
| 24 | NP_001142404 | Phosphoglycerate kinase | 207 | 42,470 | 5.64 | 16 | Energy metabolism | −1.24 | +1.06 |
| 31 | C5 × 1M7 | Lactamase_B domain-containing protein | 123 | 29,012 | 5.80 | 59 | Energy metabolism | −1.64 | −1.26 |
| 34 | P93805 | Phosphoglucomutase, cytoplasmic 2 | 205 | 63,230 | 5.47 | 41 | Energy metabolism | −1.44 | +1.29 |
| 35 | NP_001167944 | 2,3-Bisphosphoglycerate-independent phosphoglycerate mutase 1 | 312 | 60,423 | 5.47 | 35 | Energy metabolism | −1.14 | +1.02 |
| 36 | XP_002465414 | Pyruvate decarboxylase 2 | 473 | 66,424 | 5.67 | 41 | Energy metabolism | −1.73 | −1.19 |

**Table 1.** *Cont.*

| Spot No. | Accession No. | Protein Name | Score [a] | Mr [b] (kD) | pI [c] | SC [d] | Category | Fold Change (−0.5 MPa + 0 mM H₂O₂)[e] | Fold Change (−0.5 MPa + 100 mM H₂O₂)[f] |
|---|---|---|---|---|---|---|---|---|---|
| 38 | Q09EM2 | ATP synthase subunit alpha | 337 | 55,933 | 5.71 | 43 | Energy metabolism | −1.90 | −1.13 |
| 40 | XP_002467922 | 2,3-Bisphosphoglycerate-independent phosphoglycerate mutase | 520 | 61,995 | 5.83 | 44 | Energy metabolism | −1.56 | −1.24 |
| 15 | AAG23220 | Glycine-rich RNA-binding protein | 166 | 16,729 | 6.59 | 90 | Amino acid metabolism | −1.47 | −1.03 |
| 30 | C0PC62 | Uncharacterized protein | 437 | 57,479 | 5.46 | 50 | Amino acid metabolism | −1.31 | +1.32 |
| 33 | C5YW13 | Ketol–acid reductoisomerase | 338 | 63,644 | 5.88 | 45 | Amino acid metabolism | −1.12 | +1.02 |
| 39 | C0PHR4 | Adenosylhomocysteinase | 212 | 53,898 | 5.63 | 37 | Amino acid metabolism | −1.78 | −1.33 |
| 42 | AAL33589 | Methionine synthase, partial | 452 | 84,742 | 5.73 | 36 | Amino acid metabolism | −1.74 | −1.12 |
| 43 | B6TCM9 | Spermidine synthase 1 | 258 | 39,860 | 5.66 | 23 | Amino acid metabolism | −1.44 | +1.15 |
| 7 | C5WUN6 | Uncharacterized protein | 548 | 75,423 | 6.24 | 20 | Storage protein | −1.15 | +1.54 |
| 8 | C5WUN6 | Uncharacterized protein | 280 | 75,423 | 6.24 | 23 | Storage protein | −1.37 | +1.22 |
| 11 | A0A1Z5S5M9 | Uncharacterized protein | 170 | 69,817 | 7.41 | 15 | Storage protein | +1.26 | +1.43 |
| 12 | A0A1Z5S5M9 | Uncharacterized protein | 264 | 69,817 | 7.41 | 18 | Storage protein | +1.04 | +1.32 |
| 32 | C5 × 0T3 | Uncharacterized protein | 284 | 58,174 | 5.94 | 27 | Storage protein | −1.54 | +1.30 |
| 19 | NP_001130463 | Putative actin family protein isoform 1 | 257 | 41,927 | 5.24 | 51 | Cytoskelecton | −1.08 | +1.21 |
| 20 | EOY34051 | Actin-11 isoform 1 | 362 | 41,870 | 5.31 | 63 | Cytoskelecton | −1.08 | −1.21 |
| 22 | C5WT90 | Uncharacterized protein | 307 | 41,964 | 5.6 | 51 | Cell wall | −1.17 | +1.13 |
| 23 | P80607 | Probable UDP-arabinopyranose mutase 1 | 225 | 41,691 | 5.75 | 43 | Cell wall | −1.82 | +1.48 |
| 4 | C5 × 972 | Nucleoside diphosphate kinase 1 | 126 | 16,908 | 6.30 | 48 | Other protein | +1.18 | +1.40 |
| 13 | C5YTT7 | PKS_ER domain-containing protein | 509 | 38,996 | 5.8 | 78 | Other protein | −1.43 | +1.09 |
| 25 | XP_002438954 | Eukaryotic initiation factor 4A-3 | 464 | 47,222 | 5.30 | 66 | Other protein | −1.20 | +1.12 |

Note: [a] Protein match score. [b] Nominal mass. [c] Isoelectric point. [d] Sequence coverage. [e] Expression character (fold): the intensity ratio of (−0.5 MPa + 0 mM H₂O₂)/(0 MPa + 0 mM H₂O₂). [f] Expression character (fold): the intensity ratio of (−0.5 MPa + 100 mM H₂O₂)/(0 MPa + 0 mM H₂O₂).

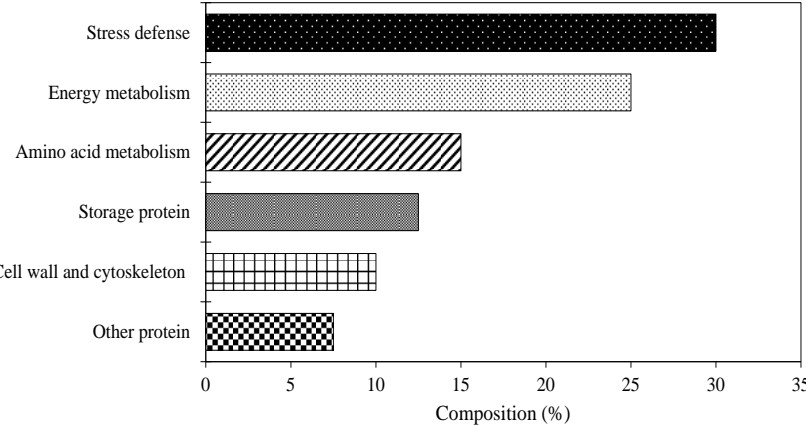

**Figure 6.** Function-based categorization of proteins expressed differentially in sorghum seed at 24 h after treatment.

Proteins involved in amino-acid metabolism (spots 15, 30, 33, 39, 42, and 43) were highly induced both in the embryo treated with 0 mM $H_2O_2$ under optimum moisture conditions and in the embryo treated with 100 mM $H_2O_2$ treatment under strong drought, but less induced in embryo treated with 0 mM $H_2O_2$ under strong drought. Storage proteins of spots 7, 8, 11, 12, and 32, albeit were not identified by mass spectrometry, were highly induced in the embryos treated with 100 mM $H_2O_2$ under strong drought. Spots 19, 20, 22, and 23 identified as proteins related to cell-wall and cytoskeletal structural proteins were highly induced both in the embryos treated with 0 mM $H_2O_2$ under optimum moisture conditions and in the embryo treated with 100 mM $H_2O_2$ under strong drought conditions. Nucleoside diphosphate kinase 1 (spot 4) was induced in the embryo treated with 100 mM $H_2O_2$ treatment under strong drought.

## 4. Discussion

As a result of artificial treatment with hydrogen peroxide, drought stress caused inhibition of seed germination, and seedling growth was ameliorated, although the effects were dependent on the hydrogen peroxide concentration. The positive effects of hydrogen peroxide on germination have been reported in the range of 1 to 100 mM in previous studies [22,42,43], our experiments also showed positive effects at this concentration range. Even though there was no significant difference in the germination percentage from 0 to 120 h after imbibition (Figure 1), the percentage was slightly higher in the seeds treated with hydrogen peroxide than those with no hydrogen peroxide treatment. According to Masondo et al. [44], moisture condition influences germination and seedling growth differently depending on the degree of water potential. Even though germination commences under drought, the embryonic axis does not produce enough energy required for growth due to a lack of water absorption. Therefore, even if germination occurs well under poor moisture conditions, it cannot grow into a healthy plant. Our results, however, showed that hydrogen peroxide treatment showed similar effects on both germination and seedling growth. Seeds treated without hydrogen peroxide showed low moisture absorption under strong drought; thus, the germination percentage was lower than those germinated under optimum moisture conditions, and the length of the radicle and plumule was also short (Figure 3). However, hydrogen peroxide treatment increased the length of the radicle and plumule under strong drought compared to the untreated control, indicating that hydrogen peroxide treatment may induce more energy production in the seed required for seedling elongation (Figure 3).

It is known that hydrogen peroxide has a positive effect on seeds because it mitigates the effects of ABA, a germination inhibitor [21]. Our results suggest that hydrogen peroxide promotes the energy metabolism required for radicle and plumule growth rather than promoting water absorption in the early stage of germination through an increase in osmotic regulators including sugars derived from storage starch [45]. Due to growth promotion of radicles and plumules, sorghum seedlings can absorb more water.

The antioxidant activity of embryos was low when hydrogen peroxide was treated artificially under drought conditions after 24 h of imbibition, although hydrogen peroxide concentration was not measured in this study (Figure 4). These results suggest that the antioxidant activity was increased due to the increase in ROS induced by drought stress, but there was no difference in the amount of water absorption by the hydrogen peroxide treatment; therefore, it is thought that hydrogen peroxide does not affect the absorption of water. Barba-Espín et al. [17] also reported that there was no significant difference between water absorption and hydrogen peroxide content in pea seeds, which could be due to ROS production via nonenzymatic reactions such as lipid peroxidation. Many reports have shown that hydrogen peroxide treatment under abiotic stress such as drought and salt improves seed germination rate, increases seedling growth, and increases intracellular antioxidant capacity [22,46–48].

Seeds and seedlings of barley [21], wheat [49], soybean [46], and cotton [50] treated with hydrogen peroxide under stress conditions were shown to alleviate stress damage by

improving the antioxidant system, stabilizing the cell membrane, reducing the amount of hydrogen peroxide in the cell, and improving the photosynthetic rate. It is assumed that reactive oxygen species were accumulated in sorghum embryos due to drought stress, and it is thought that the antioxidant capacity of seeds was increased to eliminate them. When seeds do not absorb enough water, germination and seedling growth decrease. On the other hand, hydrogen peroxide treatment induces drought resistance signaling to prevent the accumulation of reactive oxygen species up to a toxic level; therefore, it is thought that the germination and seedling growth of sorghum seeds improved even under drought conditions. Hydrogen peroxide treatment does not promote water absorption, but it is thought that the treatment is related to energy supply by respiration. Farooq et al. [51] showed that the hydrolysis of stored starch into sucrose provides essential energy for embryo growth. Although we did not analyze the changes in starch and sucrose content in seeds according to hydrogen peroxide treatment, it can be deduced that available energy was more plentiful in the hydrogen peroxide-treated seeds considering previous reports that the starch content decreased and sucrose increased in the seed under salt stress conditions by hydrogen peroxide treatment [52,53].

In the process of seed germination, respiration greatly increases, energy is produced through the glycolysis and TCA cycle, and radicle and plumule appear from the embryo to complete germination [54]. In the results of this study, most enzymes related to energy metabolism that changed in sorghum seed embryos at 24 h of imbibition were those involved in glycolysis. Among the identified proteins, fructokinase-1 (spot 16), which is involved in the process of substrate phosphorylation, was higher in the absence of hydrogen peroxide under drought conditions. It is an enzyme that converts fructose-6-phosphate to fructose-1,6-phosphate and is thought to play a role in promoting ATP consumption in drought stress. The 2,3-bisphosphoglycerate-independent phosphoglycerate mutase (spots 35 and 40) and phosphoglycerate kinase (spot 24) were increased by hydrogen peroxide treatment because they promote the production of ATP required for seedling growth. This result is also supported by the increase in ATP synthase subunit alpha (spot 38) by hydrogen peroxide treatment. ATP synthesis related to the oxidative reaction in respiration is limited in drought conditions, but it is consistent with the report of Logan et al. [55] showing that hydrogen peroxide treatment increases ATP synthesis during oxidative phosphorylation in mitochondria. When crops are under stress, they produce toxic aldehydes (e.g., methylglyoxal) [56]. Like ROS, methylglyoxal has a low intracellular concentration in the absence of environmental stress, but the intracellular concentration increases rapidly under environmental stress. The expression of a large number of proteins involved in the first step (VOC domain-containing protein; spot 2) and the second step (lactamase_B domain-containing protein; spot 31) of the methylglyoxal detoxification process seems to reduce the damage caused by intracellular methylglyoxal (Figure 5).

Hydrogen peroxide is closely related to antioxidant levels in plants. This relation can be found in a previous report [47] indicating that hydrogen peroxide treatment on mustard seeds induced low levels of oxidative stress, thereby inducing the expression of stress defense proteins. In the protein expression results of this study, antioxidant enzymes such as superoxide dismutase (SOD), glutathione S-transferase I, and 1-cys peroxiredoxin PER 1, which are known as stress defense proteins, were induced by hydrogen peroxide treatment. Increased expression of embryogenesis-abundant protein D-34 (spots 10 and 14), salt tolerance protein (spot 17), heat-shock proteins (spot 27 and 29), and chaperones (spots 29 and 41), which are highly induced under drought stress, was also confirmed, suggesting that hydrogen peroxide enhanced amelioration of stress [57,58]. The storage proteins cupin type-1 domain-containing protein (spots 7, 8, and 32) and globulin-1 S allele (spots 11 and 12) were also induced. The expression of cupin type-1 domain-containing protein (spots 7, 8, and 32) was low without hydrogen peroxide under drought conditions, while the expression of globulin-1 S allele (spots 11 and 12) was poor under the optimum water condition. The proteins (spots 15, 30, 33, 39, 42, and 43) related to amino-acid metabolism were induced by hydrogen peroxide treatment in drought conditions. This implies that

hydrogen peroxide treatment contributed to the production of free amino acids through the degradation of the stored protein.

The increased level of free amino acids coming from storage proteins may benefit the creation of new tissues during seedling formation and growth. Other proteins, such as PKS_ER domain-containing protein (spot 13), a protein involved in the synthesis of polyunsaturated fatty acids, and eukaryotic initiation factor 4A-3, were induced by hydrogen peroxide treatment under drought conditions. The probable UDP-arabinopyranose mutase 1 (spot 23) related to the synthesis of pentose complex and involved in the biosynthesis of cell wall non-cellulosic polysaccharides was reduced in the absence of hydrogen peroxide. This result suggests that hydrogen peroxide enhances physical structure through cell-wall formation. Tubulin proteins [59,60], which are highly associated with the radicle cell growth stage, were not identified in this study, but putative actin family proteins isoform 1 (spot 19) and actin-11 isoform 1 (spot 20) were identified. The increased physical strength due to induction of proteins related to the cell wall and cytoskeleton could lead to more adaptive changes against drought stress, which results in a wilting and mechanical weakening of tissues. Nucleoside diphosphate kinase 1 (spot 4), which is associated with the elongation of coleoptile, was increased by hydrogen peroxide treatment under drought. Proteome analysis suggested that hydrogen peroxide has a promoting effect on proteins related to tissue formation, tissue strength, and defense in an adverse environment under drought stress conditions rather than optimum water conditions during germination [61,62].

In agriculture, countermeasures against drought can be divided into cultivation of drought-resistant varieties, irrigation, and enhancement of soil moisture retention. This study shows that the improvement of drought resistance by treatment with low-concentration hydrogen peroxide can be used practically in agricultural fields; for example, the application of K as a fertilizer, which is effective in controlling the water status in plants, can be one of the countermeasures.

## 5. Conclusions

Sorghum seed germination and seedling growth under drought stress were increased by hydrogen peroxide treatment; however, the hydrogen peroxide treatment did not show a remarkably beneficial effect in normal conditions. Hydrogen peroxide treatment seemed to have a greater effect on the reduction of oxidative stress by ROS than the increase of water absorption in sorghum seeds under drought conditions. Proteomic changes induced with hydrogen peroxide treatment showed an increase in enzymes related to glycolysis. Proteins associated with stress defense and antioxidant enzymes were also higher in the embryos treated with hydrogen peroxide. Therefore, hydrogen peroxide treatment under drought conditions showed beneficial effects on seed germination and seedling growth by reducing oxidative stress in germinating seeds, inducing drought tolerance, and increasing enzymes related to energy metabolism.

**Author Contributions:** Conceptualization, K.E.S., e.S.H.H. and S.S.; methodology, K.E.S. and J.W.J.; software, H.R.H.; validation, K.E.S., W.J.J. and S.S.; formal analysis, K.E.S. and H.R.H.; investigation, K.E.S.; resources, S.S. and e.S.H.H.; data curation, H.R.H.; writing—original draft preparation, K.E.S. and H.R.H.; writing—review and editing, P.K., W.J.J., J.W.J. and S.S.; visualization, K.E.S.; supervision, S.S.; project administration, S.S.; funding acquisition, e.S.H.H. All authors have read and agreed to the published version of the manuscript.

**Funding:** This research was funded by the Korean Ministry of Environment, grant number 2021002270004.

**Data Availability Statement:** The data analyzed during this study are available from Sangin Shim upon reasonable request.

**Conflicts of Interest:** The authors declare no conflict of interest.

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
