# Peer review of "Hydrogen Peroxide Ameliorates the Adversities of Drought Stress during Germination and Seedling Growth in Sorghum (Sorghum bicolor L.)"

_agronomy, doi:10.3390/agronomy13020330_

Round 1

Reviewer 1 Report

The manuscript is very compact, well written and the results are presented in a very easy access way.

I just got the impression, that only selected results were presented. Was the proteomic analysis only done for the CTRL and -0.5 MPa/0 mM and -0.5MPa/100mM?

It makes totally sense to present only selected results in the main part, but I miss the supplements to check all results. If there is more data, please provide it in the supplements for the community and for transparency.

1. do you show in figure 2 and 4 selected data? is there  supplementary data where all the conditions are compared?

2. In table 1 the titles of the last two columns are missing the drought stress factor (-0.5 MPA)

Author Response

Thanks for the valuable comment
Proteomic analysis was analyzed only for treatment with 100mM which shows the effect of hydrogen peroxide well under -0.5 MPa water potential treatment, because it showed sufficient water stress in -0.5 MPa treatment.

Reviewer 2 Report

Congratulations on a well-organized, easy-to-understand and readable work. A novel approach to research hydrogen peroxide that relieves water stress during germination and seedling growth in sorghum. The topic is very interesting, the purpose and implementation of the article are clear, and the information provided is informative. The contribution of the article is clear and it will certainly become the work of researchers in this field. This article contains some technical issues that will need to be resolved before the manuscript is ready for publication.

Author Response

Thanks for the valuable comment.

Mistakes in the paper have been corrected once again.
thanks for the good advice

Reviewer 3 Report

Review Report

Title: Hydrogen Peroxide Ameliorates Water Stress during Germination and Seedling Growth in Sorghum (Sorghum bicolor)

Please modify the title as Hydrogen Peroxide Ameliorates the Adversities of Drought Stress during Germination and Seedlings Growth in Sorghum (Sorghum bicolor L.)

In this paper, the authors investigated the role of hydrogen peroxide in improving the drought tolerance in sorghum during the germination and seedling establishment by proteome analysis. The work provided novel information to understand the mechanism against drought stress with
Hydrogen Peroxide treatment. In general, this manuscript shows well design and data and merits publication in Agronomy. However, there are still some minor issues that need to be revised prior to publication. Some descriptions need to be concise and clarify.

Comments:
1. The language of the article is generally fluent, but some grammar usage should be checked

again.

2. In the abstract, Line 22 treated with hydrogen peroxide at different concentrations (0, 10, 20, 50, 100 mM etc) add the levels of concentration.

3. Add the solid conclusion at the end of the abstract section
4.The introduction of the article needs to further sort out the latest literature related to hydrogen peroxide and its role in drought tolerance during germination specifically

resistance to plants.

5. In the Instruction, there is little reports which focused on drought resistance of sorghum, please add it.

6. Why author used only one variety specifically “(cv. Hwangeumchal)”
7. How much volume of each concentration of hydrogen peroxide solution was used in the medium for each petri dish?

8. Line 92: author mentioned that All treatments combination were replicated four times but in figures authors mentioned that All values are means ± SE of three replicates which one is correct  

9. Please expand the statistical section, write in detail SE (Standard error), detail of turkeys test and ANOVA process and level of significance

10. I think the level of significance in control condition must be 1% instead of 5%

11. Results and discussion section of the article is weak, authors mainly focused on their results but they did not discuss them according to international standards. Moreover the writing style of results and discussion section is also ambiguous, with long and weak sentences and in a repetitive way. I am not convinced with the way of discussion of the authors, in its current form it cannot be accepted in agronomy. I will recommend a thorough revision of this section.

12. The reference of the article needs to be checked, revised and formatted.

Author Response

Thanks for your valuable comment.
1. The title has been modified.
2. Verbosity in expression has been corrected.
3. The level of concentration was indicated in the abstract.
4. Added a concise conclusion to the abstract.
5. Added information about drought damage in sorghum.
6. Added the recent results about germination and the effects of hydrogen peroxide in sorghum.
7. This study was limited to the most cultivated varieties in Korea rather than knowing the differences between varieties.
8. Added the media volume (5 mL per petri dish).
9. Corrected with four replicates.
10. Added statistical significance level.
11. The level of significance was checked.
12. Revised the discussion.
13. References have been revised.

Thanks for the good advice